# Potential Biomarkers of Skin Melanoma Resistance to Targeted Therapy—Present State and Perspectives

**DOI:** 10.3390/cancers14092315

**Published:** 2022-05-06

**Authors:** Magdalena Olbryt

**Affiliations:** Center for Translational Research and Molecular Biology of Cancer, Maria Sklodowska-Curie National Research Institute of Oncology, Gliwice Branch, 44-102 Gliwice, Poland; magdalena.olbryt@io.gliwice.pl

**Keywords:** melanoma, BRAF, targeted therapy, resistance, predictive biomarker, genetic test

## Abstract

**Simple Summary:**

Around 5–10% of advanced melanoma patients progress early on anti-BRAF targeted therapy and 20–30% respond only with the stabilization of the disease. Presumably, these patients could benefit more from first-line immunotherapy. Resistance to BRAF/MEK inhibitors is generated by genetic and non-genetic factors inherent to a tumor or acquired during therapy. Some of them are well documented as a cause of treatment failure. They are potential predictive markers that could improve patients’ selection for both standard and also alternative therapy as some of them have therapeutic potential. Here, a summary of the most promising predictive and therapeutic targets is presented. This up-to-date knowledge may be useful for further study on implementing more accurate genetic/molecular tests in melanoma treatment.

**Abstract:**

Melanoma is the most aggressive skin cancer, the number of which is increasing worldwide every year. It is completely curable in its early stage and fatal when spread to distant organs. In addition to new therapeutic strategies, biomarkers are an important element in the successful fight against this cancer. At present, biomarkers are mainly used in diagnostics. Some biological indicators also allow the estimation of the patient’s prognosis. Still, predictive markers are underrepresented in clinics. Currently, the only such indicator is the presence of the V600E mutation in the BRAF gene in cancer cells, which qualifies the patient for therapy with inhibitors of the MAPK pathway. The identification of response markers is particularly important given primary and acquired resistance to targeted therapies. Reliable predictive tests would enable the selection of patients who would have the best chance of benefiting from treatment. Here, up-to-date knowledge about the most promising genetic and non-genetic resistance-related factors is described. These are alterations in MAPK, PI3K/AKT, and RB signaling pathways, e.g., due to mutations in *NRAS*, *RAC1*, *MAP2K1*, *MAP2K2*, and *NF1*, but also other changes activating these pathways, such as the overexpression of HGF or EGFR. Most of them are also potential therapeutic targets and this issue is also addressed here.

## 1. Introduction

According to the National Institute of Health (NIH), a biomarker is a feature that can be objectively measured and estimated as an indicator of physiological and pathogenic biologic processes as well as an indicator of response to therapy [1]. In oncology, biomarkers are commonly used for the diagnosis of cancer, evaluation of disease stage and prognosis, as well as prediction and monitoring of treatment outcome [2]. Biomarkers differ in their nature. We can distinguish genetic markers (e.g., changes in the DNA sequence), biochemical markers (e.g., proteins and other substances in plasma), molecular markers (e.g., gene expression signatures), or cytological markers (e.g., lymphocyte infiltration). Concerning melanoma, diagnostic markers are mainly used to differentiate between benign and malignant lesions and to distinguish melanoma from other types of neoplasms, e.g., sarcoma. These are proteins that are less (e.g., S100 family) or more (e.g., MELAN-A/MART-1) specific for melanoma cells [3]. Some parameters of primary tumors are used to estimate the prognosis, e.g., Breslow thickness, ulceration, or mitotic index. The prognostic value is also well established for the level of lactate dehydrogenase (LDH) in the blood of patients with disseminated disease. A high level of this enzyme correlates with shorter overall survival [4]. The least numerous group are predictive markers.

Until 2011, there was no effective drug to prolong the survival of patients with advanced and unresectable cutaneous melanoma. Today, two main new therapeutic strategies are routinely used: molecularly targeted therapy (dabrafenib, vemurafenib, encorafenib, trametinib, cobimetinib, binimetinib) and immunotherapy (pembrolizumab, nivolumab, ipilimumab). Additionally, in the case of the presence of mutations in genes other than *BRAF* (B-Raf proto-oncogene, serine/threonine kinase), alternative targeted therapy may be considered, e.g., with imatinib, when a mutation in the *c-KIT* gene is present. It is also possible to treat injectable melanoma with the genetically modified oncolytic virus (talimogene laherparepvec) [5]. Adjuvant therapy with kinase inhibitors (dabrafenib and trametinib) and immunotherapy (pembrolizumab, nivolumab, ipilimumab) for high-risk melanoma are also registered [6]. Although many patients take advantage of these new therapies, some patients do not respond to the treatment, both targeted and immunological. The development of reliable markers of response would allow for better personalization of the treatment and consequently would lead to improved patient survival and lower costs of patient care.

## 2. Molecularly Targeted Therapy of Skin Melanoma

In 2011, the FDA approved the first targeted drug for the treatment of advanced melanoma, the BRAF protein inhibitor vemurafenib. BRAF is a kinase that belongs to the BRAF-MEK-ERK signaling pathway (MAPK pathways) and is mutated in approximately 60% of skin melanoma cases [7]. The mutant form is nearly five hundred times more active than the wild form of the protein, leading to continuous activation of the MAP kinase signaling pathway and unlimited tumor growth [8]. Detection of BRAFV600 mutation (with V600E being the most frequent aberration and V600K being the second one) in melanoma cells qualifies the patient for targeted therapy with BRAF, MEK inhibitors (BRAFi/MEKi).

Currently, there are three BRAF (vemurafenib, dabrafenib, encorafenib) and three MEK inhibitors (trametinib, kobimetinib, binimetinib), as well as their combinations approved for advanced melanoma treatment. The inhibitors selectively bind to the mutated kinases and inhibit their activity leading to decreased cell proliferation, cell cycle arrest, and induction of apoptosis [9]. Therapy with these drugs is well tolerated and improves the survival of most patients with advanced melanoma [10]. However, the primary and acquired resistance is a significant limitation of this therapeutic strategy and occurs in nearly 100% treated with monotherapy [11]. Primary resistance occurs when the patient, despite a diagnosed mutation in the *BRAF* oncogene, does not benefit from targeted therapy. It is estimated that 5% to 20% of patients do not respond satisfactorily to treatment despite the presence of mutations in the BRAF gene. The problem concerns approximately 5–15% of patients whose disease progression is observed at the first imaging examination after the initiation of monotherapy [12,13]. In the case of combined therapy (anti-BRAF and anti-MEK), primary resistance is observed in a few percent of patients (disease progression), while 20–30% of treated patients only benefit with disease stabilization [14,15]. The response to the treatment is a favorable prognostic factor affecting the time to progression and overall survival [16]. Unfortunately, if the course of the disease during molecularly targeted treatment is dynamic, the proportion of patients may not qualify for second-line immunotherapy due to poor general conditions. Palliative care remains the only option for such patients.

The possibility of estimating the probability of response to targeted therapy before treatment would allow for a better selection of therapy and thus increase the effectiveness of the treatment of advanced skin melanoma. Therefore, it is important to identify markers of primary or early resistance to this therapy. It seems especially useful as many genetic and molecular mediators of tumor resistance are also potential therapeutic targets.

## 3. Where Does Resistance to Targeted Therapy in Melanoma Cells Come from?

The main cause of primary resistance of melanoma to targeted therapy is the high heterogeneity and plasticity of this tumor, characterized by the presence of many cell clones within the tumor [17,18]. The formation of multiple clones is favored by the presence of numerous UV-induced mutations in melanoma cells [19], with an average number of about 50 mutations per million base pairs of DNA [20]. By comparison, the frequency of mutations in breast cancer is several per million bp. [19]. In addition to the genetic variability of melanoma, resistance to BRAF inhibitors can also be generated by the microenvironment and epigenetic changes [21]. Therefore, potential predictive markers can be divided into two types: genetic and non-genetic described below, presented in Table 1 and Figure 1.

## 4. Potential Genetic Predictive Markers in the Targeted Therapy of Melanoma

Resistance to targeted therapy results mainly from the reactivation of the MAPK pathway or activation of an alternative pathway regulating cell division, the PI3K/AKT pathway [52]. So far, several genetic changes which contribute to primary resistance have been identified. These are mutations in the following genes: *RAC1*, *MAP2K1*, *MAP2K2*, *NF1*, and *PTEN*. The potential predictive value may also have mutations in the *RB1*, *PIK3CA*, *MYC*, *CDKN2A* genes, and to a lesser extent *CDK4*, *NRAS*, and *BRAF* amplification [22,24,28].

### 4.1. RAC1

RAC1 is a signaling molecule that belongs to the RAS superfamily of small GTP-binding proteins. It plays an important role in key features of malignant tumors such as proliferation, metastasis, and resistance to therapies [53]. Mutations in RAC1 occur in skin melanoma with a frequency of 5–9% [54,55]. The most common mutation in this gene is a change in codon 29 (P29S). It is a “gain of function” mutation, which causes a constant activation of the MAPK signaling pathway, increasing the proliferation of melanoma cells [56], and leading to cell resistance to inhibitors of the kinase pathway MAP. It has been shown that cells with this mutation are several times more resistant to inhibitors of BRAF and MEK kinases compared to cells without this mutation. A similar effect was also observed in vivo in a mouse model [25,57]. *RAC1*^P29^ mutations are identified in tumors of patients with early resistance [22] and those treated with combined targeted therapy, whose time to progression (TTP) was shorter than the median TTP for this group of patients [24,29]. Analysis of TCGA data revealed significantly shorter relapse-free survival of patients with *RAC1*^P29S^ mutation in comparison to those with wild-type RAC1 [57]. The above results and the fact that the *RAC1*^P29^ mutation is identified in almost one in ten patients suggest that this change should be a part of a potential predictive panel for skin-melanoma-targeted therapy [58]. Importantly, inhibitors for the RAC effector p21-activated kinase (PAK) might impede oncogenic signaling from mutated *RAC1* [26], which makes this gene a potential therapeutic target.

### 4.2. MAP2K1 and MAP2K2

The reactivation of the MAPK pathway during therapy with BRAF and MEK kinase inhibitors is also the result of mutations in the *MAP2K1* and *MAP2K2* genes. The proteins encoded by these genes (MEK1 and MEK2) are the major kinases of the Ras-Raf-MEK-ERK pathway activated by the BRAF oncogene. Activating mutations in these genes may not only promote the development of melanoma but also reduce the effectiveness of BRAF/MEK inhibitors as a result of activation of the MAPK pathway despite the inhibition of BRAF kinase [59]. Changes in these genes are identified in samples of melanomas of patients with a worse response to targeted therapy (time to progression (TTP) < 4 months; [24] and in patients with primary and acquired resistance to BRAF inhibitors [22,29]. In vitro, functional studies have shown that selected mutations in the *MAP2K2* and *MAP2K1* genes can generate resistance to MAPK kinase inhibitors [22,59,60]. It seems that the effect of mutations in these genes may be different in the case of monotherapy and combination therapy (combined BRAF and MEK inhibitors); for example, *MAP2K1*^C121S^ mutation reduces the effectiveness of the BRAF [59,60] and MEK [60] inhibitors used separately, but it seems to not affect the combined effect of both inhibitors [24]. Further research is needed to clarify this point. The results so far suggest that some mutations in the *MAP2K1* and *MAP2K2* genes may contribute to the primary resistance of melanoma cells to targeted therapy and should be further investigated both functionally and in clinics as a part of the genetic predictive panel. The most common mutations related to melanoma resistance are listed in Table 1.

### 4.3. NF1

Primary resistance resulting from the simultaneous reactivation of the MAPK pathway and activation of the PI3K/AKT pathway may be generated by a mutation in the *NF1* (neurofibromin 1) gene [61]. Mutations in the *NF1* gene leading to the inactivation of the encoded protein occur in melanoma with a frequency of 12–30% [62] and are the third most frequent mutation in melanoma (after *BRAF* and *NRAS*). The NF1 protein is a tumor suppressor that inhibits the activity of the NRAS oncogene. Lack of NF1 leads to increased activity of the NRAS protein, and thus the activation of both the MAPK pathway and the PI3K/AKT pathways. Therefore, the presence of a mutation in the *NF1* gene together with a mutation in the *BRAF* gene may limit the effectiveness of BRAF/MEK inhibitors. It has been shown that the inhibition of NF1 expression reduces the sensitivity of cells to the BRAF inhibitor by up to 30-fold and to the MEK inhibitor several-fold [23]. Additionally, knockdown of this gene renders melanoma cells resistant to vemurafenib [63]. Mutations in the NF1 gene are also present in the tumor DNA of patients primarily resistant to BRAF inhibitors [23] and patients with early or acquired resistance to these drugs [22,28]. It seems that due to the relatively frequent occurrence of *NF1* mutations in melanoma and the fact that the inactivation of this gene reduces the sensitivity of melanoma cells to both types of MAP kinase inhibitors (anti-BRAF and anti-MEK), genetic changes in this gene are a promising predictive marker in targeted therapy of skin melanoma. A limitation of such testing is the size of the gene (282,751 bp; mRNA—about 12,000 bp) and the lack of hot-spot mutations, which in the case of developing a predictive test will involve the sequencing of the entire gene and thus increased costs of the test.

### 4.4. PTEN

PTEN is a tumor suppressor protein that inhibits the activity of the PI3K/AKT pathway. Changes in the *PTEN* gene leading to protein inactivation occur in approximately 30–35% of melanoma cases [54]. When the pro-survival action of the MAPK pathway is inhibited by BRAF/MEK inhibitors, the PI3K/AKT pathway can take over this function, thereby reducing the effectiveness of the therapy. Genetic analyses have shown that *PTEN* mutations are more common in the tumors of patients who did not respond (primary resistance) or responded poorly (PFS < 6 months) compared to those who benefited from this therapy [22,24,64]. Additionally, a shorter progression-free time was observed in patients with a dysfunctional *PTEN* gene treated with a BRAF inhibitor compared to patients with a wild-type PTEN [64,65]. Some results of in vitro studies suggest that cells with the mutated PTEN gene are less sensitive to BRAF [66] or MEK [67] inhibitors. The potential mechanism involves the upregulation of PERK (EIF2AK3), inhibition of which restores the sensitivity of PTEN-impaired melanoma cells to BRAF inhibitor [68]. It has also been shown that cells lacking a functional *PTEN* gene are less sensitive to vemurafenib-induced apoptosis [69]. However, in this study, no difference in response to a BRAF inhibitor was observed between PTEN+ and PTEN− cells. Not all patients with impaired *PTEN* genes show a worse response to therapy with BRAF/MEK inhibitors [22,29]. Therefore, it seems that the lack of a functional PTEN protein may modulate the response to BRAF/MEK therapy, but it is not a sufficient genetic event to generate resistance to these drugs [61]. It is also possible that the effect of PTEN loss may be more relevant for monotherapy, while less important for combined treatment as was shown in the coBRIM study. PTEN loss was associated with shorter PFS only in patients treated with vemurafenib, but not cobimetinib and vemurafenib [70]. Nevertheless, the genetic analysis of this gene should certainly be included in a potential predictive test as it may suggest targeting PI3K/AKT pathway as an alternative to MAPK inhibition in case of resistance development.

### 4.5. CDKN2A

The impaired functioning of the *CDKN2A* gene may also have a potential impact on the sensitivity of melanoma cells to targeted therapy. It encodes two (frameshift) proteins: p16INK4a and ARF. Genetic aberrations in *CDKN2A* are identified in about 40–60% of melanomas [54,71], and almost all lines derived from tumors have a genetically impaired CDKN2A/B-ARF pathway [72]. Changes in this gene, especially deletions, most often lead to a simultaneous impairment of both the p53 pathway (via the ARF protein) and the RB pathway (via the p16INK4a protein). Hereditary mutation in *CDKN2A* strongly predisposes to skin melanoma and is diagnosed in about 25–50% of families with the aggregation of this cancer [73]. A mutation in the *CDKN2A* gene is identified in melanomas of patients whose time to progression was shorter than the median for patients treated with BRAF inhibitors [21,24,25]. Forschner [30] et al. described a case of a patient with melanoma harboring a homozygous deletion of *CDKN2A*, *NRAS* mutation, as well as amplification of *CCNE1* and *CDK6*. The tumor was refractory to both combined immunotherapy and targeted therapy with BRAF/MEK inhibitors; however, it responded to combined MEK and CDK4/6 inhibition. Similar results were obtained in PDX models of BRAF-V600E-mutant melanoma refractory to BRAFi/MEKi therapy which harbored mutations in *NRAS* and *CDKN2A* [31]. *CDKN2A* mutations are also responsible for 11% of MAPK-reactivating mechanisms of resistance among disease-progressive melanomas described by Shi [74] et al., while a lower copy number of *CDKN2A* was associated with shorter progression-free survival of patients treated with dabrafenib [65]. The aforementioned data suggest that *CDKN2A* mutations may contribute to melanoma resistance; however, to my knowledge, there are no in vitro data confirming the influence of this gene dysfunction and individual mutations on the sensitivity of melanoma cells to BRAF/MEK inhibitors.

### 4.6. Other Resistance-Related Genes: PIK3CA, MYC, CDK4, MAP3K8

Mutations in *PIK3CA* [22,24,28,29] are identified in pre-treatment tumor samples of patients who respond poorly to targeted therapy, but their role in generating resistance has not yet been verified in melanoma functional studies. *PIK3CA*^E545K^ was identified as a mutation that pre-existed in rare melanoma subpopulations before therapy and was proved to render resistance to MEK/CDK4 inhibitors [75]. Interestingly, *PIK3CA*^H1047R^ mutation was also shown to generate resistance of *BRAF*^V600E^ thyroid cancer to BRAF inhibition in a murine model [76].

The role of the *CDK4* gene in generating resistance also requires investigation. In vitro studies indicate that mutations in the *CDK4* do not per se induce resistance to BRAF/MEK inhibitors, but rather enhance resistance generated by increased expression of cyclin D1 (CCND1) [77]. Our genetic analysis, on the other hand, suggests that the *CDK4*^R24^ mutation may have an impact on the response to targeted therapy. Out of 37 analyzed cases, this mutation was present in three patients treated either with mono or combined therapy whose time to progression (TTP) was shorter than the median TTP for this group of patients [29].

There is also a rationale for the role of MYC in melanoma resistance to BRAF/MEK inhibitors. MYC has been proved to be a convergent downstream effector of resistance in melanoma caused by the reactivation of such pathways as ERK, PI3K, NOTCH1, and others. Its overexpression is related to both intrinsic and acquired resistance of melanoma cells to BRAF/MEK inhibitors [33]. In our study, we have detected VUS and amplification of this oncogene in samples of two patients with a very short time to progression (<3 months) [29]. This gene is definitely worth further study and inclusion in the predictive panel.

In the case of the *MAP3K8* (COT) gene, functional studies have shown that its increased expression (as a result of amplification) causes the reactivation of the MAPK pathway and resistance of melanoma cells to MAPK inhibitors [32]. Increased expression of this protein is also present in some clinical specimens of resistant melanomas [32,78]. Interestingly, rearrangements and amplification of the *MAP3K8* gene leading to increased levels of the truncated, active form of this protein are detected in approximately 15% of melanomas without driver mutations, such as *BRAF*, *NRAS*, or *NF1* [79].

All the above-mentioned genes regulate the key signaling pathways in the development of melanoma, namely the MAP kinase pathway (*NRAS*, *MEK1*, *MEK2*, *MAP3K8*), the PI3K/AKT pathway (*PTEN*, *RAC1*, *PIK3CA*) and the RB pathway (*CDKN2A*). Mutations in the major genes of these pathways can lead to the reactivation of the MAPK pathway, activation of the alternative PI3K/AKT pathway, or increased proliferation of cells with impaired cell cycle control. These are the main mechanisms of melanoma cell resistance to targeted therapies but are not the only ones. In some cases, mutations in other genes, such as *MITF*, may play a key role. MITF is a transcription factor that regulates the expression of genes specific to melanocytes, which code for proteins involved in the production of melanin. The MITF pathway is dysregulated in approximately 15% of melanoma cases, and *MITF* amplification occurs in 10% of primary melanoma and 15% of metastatic diseases [80]. Mutations in this gene are also detected in patient samples both before therapy [22,29] and samples on-progression [22]. It has been shown that increased expression of the MITF oncogene reduces the sensitivity of melanoma cells to targeted therapy by several dozen times compared to cells with wild-type protein [22], although there are also contradictory results, which are described in the next chapter.

Another ‘melanoma’ gene potentially responsible for resistance to targeted therapies is the *GNAQ*. It belongs to the Gα1/Q pathway, deregulation of which is responsible for the development of eye melanoma [81]. It may also modulate the response to MAP kinase inhibitors. Turajlic [34] et al. described a case of a patient who experienced a very rapid progression during targeted therapy of skin melanoma presumably due to the simultaneous presence of mutations in *GNAQ* and *PTEN* genes.

### 4.7. The Role of BRAF and NRAS in the Acquisition of Melanoma Cells’ Resistance to MAPK Inhibitors

*NRAS* mutations and *BRAF* amplification are also often identified in genetic analysis of melanoma samples. In most cases, these changes are detected both in neoplasms that progressed during therapy [36] and melanoma samples collected before treatment from patients who responded to therapy [24]. Thus, it seems that both genetic alterations are more responsible for acquired resistance than for primary resistance. They may occur together with other genetic changes that generate resistance. It is estimated that in approximately 20% of cases, secondary resistance to BRAF/MEK inhibitors is multi-causal [11].

Alternative BRAF splicing is also associated with resistance to BRAFi/MEKi. Melanoma cells with *BRAF* deprived of exons 4–8 are selected during therapy, which leads to tumor resistance to vemurafenib [37]. The potential mechanism of BRAF splicing variant-related resistance is the enhanced association of *BRAF* ΔEx with its substrate MEK [82]. Splicing variants of the *BRAF* oncogene have been identified in the samples of patients with acquired resistance, while such forms were not present in the primary resistant samples [37,38]. This suggests that this mechanism of resistance is responsible for acquired insensitivity rather than for inherent resistance to the treatment.

### 4.8. Multi-Genetic Cause of Melanoma Resistance to Targeted Therapy

In most melanoma cases, there is more than one mutation [54]. The simultaneous presence of mutations in *NRAS* and *MAP2K1* genes, *NRAS* mutation and the amplification of *BRAF*, as well as the presence of two mutations in the *NRAS* gene (Q61R, T58I) have been described concerning treatment resistance [22]. Additionally, the co-occurrence of *GANQ* and *PTEN* mutations was described in a case report of a patient with progressed disease [34]. In our study, we have identified more than one genetic alteration with a potential impact on the response to therapy in several patients, e.g., in a patient refractory to therapy, a mutation in the *CDK4* gene and amplification of the *MYC* oncogene were present [29]. The significance of the presence of co-occurring mutations to therapy resistance requires further research, but the foregoing data suggest that combinations of mutations in *MAP2K1*, *MAP2K2*, *RAC1*, *NF1*, and mutated BRAF before therapy may be unfavorable for targeted therapy with BRAF/MEK inhibitors. Additionally, co-occurrence of other alterations namely *PTEN* deletions, *MYC* amplification, or *CDK4* mutations may contribute to resistance to targeted therapy with BRAFi/MEKi, but the data are too scarce and inconclusive.

## 5. Potential Non-Genetic Predictive Markers in Targeted Therapy of Melanoma

Primary resistance is also influenced by factors regulating the gene expression and those related to the tumor microenvironment that modulates the response to therapy through molecules secreted by non-cancerous tumor cells and the state of hypoxia. In the context of the search for predictive biomarkers, the most useful are proteins that modulate the response to targeted therapy, and their expression changes depending on the degree of sensitivity/resistance of tumor cells to BRAF/MEK inhibitors. The most promising proteins are described below.

### 5.1. Hepatocyte Growth Factor (HGF)

One of the proteins secreted by tumor-associated stromal cells, the activity of which may promote the resistance of melanoma cells to inhibitors of the MAP kinase pathway, is the hepatocyte growth factor [40]. HGF is a ligand for the c-MET tyrosine kinase receptor. It is secreted by fibroblasts inhabiting the neoplastic tumor and, together with other proteins, determines the pro-neoplastic properties of these fibroblasts [83]. The HGF/c-MET pathway plays a key role in the proliferation, survival, metastasis, and resistance of melanoma cells to MAP kinase inhibitors [84]. It has been shown that HGF secreted by fibroblasts causes the primary insensitivity of melanoma cells to PLX4720 (BRAF inhibitor) as a result of the simultaneous activation of the MAPK and PI3K/AKT pathways. Increased expression of this protein in the extracellular matrix of neoplastic lesions correlates with resistance to therapies [40]. HGF rescues BRAF-mutated cell lines from growth inhibition caused by both mono and combination treatment with BRAF inhibitors and this effect is attenuated by the inhibition of MET signaling [85]. HGF also induces resistance to trametinib in uveal melanoma cells [86] and dasatinib (KIT inhibitor) in acral melanoma [87]. A correlation was also observed between the level of HGF in the blood and the progression-free time (PFS) and overall survival (OS) of patients treated with vemurafenib [88]. HGF is secreted not only by fibroblasts but also by melanoma cells stressed with such factors as drugs, and reduced glucose or oxygen concentration (hypoxia). Together with other proteins, it is responsible for the phenomenon of the so-called tolerance to stress factor-induced drugs (IDTCs, induced drug-tolerant cells), which is reversible after drug holidays [89].

It has been shown that the low oxygen concentration decreases the sensitivity of melanoma cells to vemurafenib several times, which is probably due to the increased activation of the PI3K/AKT and HGF/c-MET pathways [41]. The suggestion that the activation of the HGF/MET pathway under the influence of various factors (hypoxia, fibroblasts) may contribute to a lower sensitivity of melanoma cells to targeted therapies is also confirmed by the observations of an increased level of HGF (mRNA) expression in samples progressing during the therapy compared to samples before therapy [41]. More about the role of the HGF/MET pathway in acquiring resistance to targeted drugs in the treatment of lung and colon cancer, as well as melanoma, can be found in the review by Della Corte [42] et al. The aforementioned results argue for further evaluation of this protein as a predictive marker of melanoma response to targeted therapy.

### 5.2. Tyrosine Kinases Receptors

Activation of receptor tyrosine kinases (RTKs) and increased expression of growth factors may lead to the stimulation of the MAP kinase pathway and/or the PI3K/AKT pathway and is, therefore, one of the mechanisms of resistance to targeted therapy with BRAF/MEK inhibitors. In melanoma cells, the expression of receptor tyrosine kinases such as EGFR, PDGFRB, and ERBB3 increases upon exposition to BRAF/MEK inhibitors [44].

High expression of EGFR causes primary resistance to vemurafenib in colorectal cancers carrying BRAF oncogene mutations [90]. It cannot be ruled out that also in melanoma, strong expression of this receptor may contribute to early resistance to targeted drugs. In a study by Sun [44] et al., 6 out of 16 patients were characterized by the increased expression of EGFR in tumor samples obtained after targeted treatment. The authors suggest that EGFR expression causes adaptive tolerance to the drug, which is likely to disappear upon discontinuation of the drug. Increased expression of EGFR together with urokinase receptor (uPAR) was also observed in relapsed patients [91]. BRAF-mutant melanoma cells with increased expression of EGFR are less sensitive to BRAF inhibitors [92] and elevated expression of EGFR together with NGFR characterizes pre-resistant melanoma cells, which are selected during therapy and lead to treatment failure [93]. Additionally, the depletion of a negative regulator of EGFR protein degradation (ACK1) results in the accumulation of EGFR and melanoma cells which confers resistance to targeted therapy [94]. The involvement of proteins regulating EGFR in melanoma resistance indirectly proves the role of this receptor in response to targeted therapies [95,96]. It is plausible that EGFR mutations that occur in a certain percentage of melanomas [97,98] may contribute to primary resistance. Genetic studies on a large number of patients would allow the verification of this thesis. On the other hand, due to the significant role of epigenetics in the regulation of EGFR, evaluation of EGFR expression in tumor samples could be more informative than genetic testing. Previous studies on colon cancers suggest the usefulness of EGFR status in predicting response to BRAF inhibitors [99,100].

Insulin-like growth factor receptor 1 (IGF1R) is also involved in generating resistance of melanoma cells to inhibitors of the MAPK pathway. This protein is involved in the transformation of melanocytes [101] and the development of melanoma [102]. IGF1R is stabilized in melanoma cells by the activation of the BRAF oncogene, PTEN suppressor, or by contact with cancer-associated fibroblasts, which leads to its increased expression. Cells with IGF1R overexpression are more resistant to vemurafenib, and the downregulation of this protein leads to the sensitization of melanoma cells to the drug by inducing apoptosis [103]. Activation of the IGF1R–MEK5–Erk5 pathway was also proved to cause resistance to double treatment with BRAF/MEK inhibitors in melanoma cells [104]. Treating resistant melanoma cells with IGF1R inhibitors restores their BRAFi sensitivity [104,105,106], and a combination of BRAF/MEK inhibitors with IGF1R inhibitors significantly reduces melanoma cell growth in vitro and in vivo [107]. Increased expression of IGF1R protein was detected in melanoma samples of relapsed patients [47] suggesting its involvement in acquired resistance. It would be interesting to study the expression of this protein in pre-treatment melanoma samples in patients with primary resistance to BRAFi/MEKi. Possibly, insulin-like growth factor 1 (IGF1) also contributes to the resistance of melanoma cells to BRAF inhibitors. It is one of the proteins secreted by melanoma cells sensitive to the BRAF inhibitor during treatment. This unique therapy-induced secretome activates the PI3K/AKT pathway in resistant cells, leading to their growth and treatment failure [108].

PDGFB (PDGFRB) and PDGFA (PDGFRA) receptors may also play a certain role in generating resistance to MAPK inhibitors. These are tyrosine kinases that act as receptors for the PDGF family of growth factors that regulate such processes as embryonic development or wound healing [109]. They play an important role in the process of carcinogenesis, as well as the resistance of various cancers to therapies [110]. It has been shown that increased PDGFRB expression in melanoma cells makes them resistant to the BRAF inhibitor. High expression of this protein is also present in some treatment-resistant metastatic lesions [45,111]. Similar results were obtained for PDGFRA. Increased expression of this protein causes cell resistance to vemurafenib, while the inhibition of PDGFRA activity restores cell sensitivity to this inhibitor [46].

Functional studies of the effects of tyrosine kinase receptors on the sensitivity of melanoma cells to BRAF/MEK inhibitors indicate that these proteins may be involved in generating resistance to these drugs. However, further studies are needed to assess the potential relationship between the expression of the aforementioned proteins and the response to the targeted therapies concerning the usefulness of these proteins as predictive biomarkers.

### 5.3. Other Proteins Involved in the Generation of Resistance to Targeted Therapy in Melanoma

One of the proteins involved in the generation of resistance to BRAF/MEK inhibitors is the MITF oncogene. MITF is the major regulator of the plasticity of the melanoma cell phenotype. Its increased level induces cell proliferation, while the decrease in expression gives a signal to a more invasive phenotype regulated by the proteins WNT5A and NFĸB [49]. Lack of MITF expression generates not only acquired resistance but also significantly reduces the sensitivity to BRAF and MEK inhibitors of cells that have not been previously treated with these compounds. Cell resistance correlates with increased expression of AXL, EGFR, or PDGFRB tyrosine kinase receptors. It is suggested that a low MITF/AXL expression ratio may be a predictor of response to targeted therapy. Low MITF expression may also modulate the response to MAPK inhibitors in cooperation with NFĸB. Konieczkowski [50] et al. observed that low MITF expression and high NFĸB characterized melanoma cells resistant to MAPK inhibitors. Reduced expression of MITF during the development of resistance was also observed in the majority of melanoma cell populations derived from patients [112]. On the other hand, very high expression of MITF may also result in a decrease in the sensitivity of cells to MEK inhibitors [22,51]. The role of interplay between MITF, AXL, and other factors in the tumor microenvironment was comprehensively described in the review paper by Arozarena [113] et al. The influence of MITF on melanoma cell resistance is not fully understood [18]; therefore, for now, the use of this protein expression as a predictive marker is questionable. Assessing this protein together with other molecules, namely AXL and NFĸB, seems more rational. The issue is even more complicated by the fact that MITF regulates the genetic phenotype switching program that governs drug addiction [114]. The dependency of tumors on therapeutic drugs to which they have acquired resistance is a fascinating phenomenon [115]; however, it is beyond the scope of this review paper.

Other proteins that can modulate resistance to BRAF/MEK inhibitors are RIP1 [116] and FRA1 (FOSL1) [108]. RIP1 is a serine-threonine kinase involved in the regulation of inflammation and death. This protein has been shown to inhibit apoptosis of cells treated with MAPK pathway inhibitors by activating the NFĸB pathway, which, according to the authors, may be responsible for both primary and acquired resistance of cells to the above-mentioned inhibitors [116]. FRA1, on the other hand, regulates the secretion of compounds (TIS, therapy-induced secretome) in cells sensitive to BRAF/MEK inhibitors, which stimulate the growth of resistant cells [108]. This defensive stress response promotes the growth of cells resistant to the therapy and also helps circulating tumor cells to repopulate regressed tumors. Decreased FRA1 expression was observed in cell lines sensitive to targeted therapy (both melanoma and lung cancer) and patients’ tumors before treatment. Functional tests have confirmed the key role of this molecule in inducing resistance to MAPK inhibitors [108] as well as the addiction of melanoma cells to BRAFi/MEKi [114].

Recent studies suggest that, paradoxically, functional TP53 may contribute to melanoma resistance to targeted therapy. TP53 induces a therapy-resistant phenotype of slow-cycling cells in response to stress (e.g., therapy) and its inhibition promotes sensitivity to BRAF/MEK inhibitors [117]. The opposite role is suggested for other tumor suppressors RASSF1A. It is frequently silenced in melanoma cells and as a hub of many signaling pathways has the potential to modulate resistance to targeted therapy [118]. Other proteins with potential predictive capacity involve those regulating metabolism. Drug-tolerant melanoma cells acquire dependencies on fatty acid metabolism [119] and some proteins regulating this process are overexpressed in non-responding patients, e.g., PPAR [120]. Additionally, increased activity and expression of ABL1/2 kinases were proved to induce resistance to BRAF/MEK inhibitors and ABL kinase inhibitor, nilotinib caused the prolonged regression of resistant tumors [121]. STAT3–PAX3 signaling is also involved in generating the resistance of melanoma cells to BRAF inhibitors [122]. However, data on the aforementioned proteins and their role in resistance generation are scarce. Thus, they are not considered predictive markers, yet.

## 6. Conclusions

The existing clinical data show that patients with advanced and dynamic diseases should be treated with kinase inhibitors first, and immunotherapy should be considered only in the case of progression. Unfortunately, some patients after the failure of anti-BRAF therapy do not qualify for immunotherapy due to their poor general condition.

Therefore, apart from mutations in the *BRAF* gene, additional predictive factors are needed, that will allow the selection of patients for appropriate therapy. There are genetic and non-genetic mechanisms of primary and acquired resistance to BRAF and MEK kinase inhibitors (Figure 1). Some of them are common to primary and acquired resistance, such as mutations in the *MEK1*, *PTEN*, *CDKN2A*, or *PIK3CA* genes, while some seem to be specific to primary (*RAC1*, *MEK1* mutations) or acquired (*BRAF* amplification, *NRAS* mutations).

From the clinical point of view, those that would allow better qualification of patients for targeted therapy and at the same time indicate potential new therapeutic targets in the treatment of advanced melanoma are important. The genes with the highest predictive potential are highlighted in Table 1. The table contains the most studied genetic and non-genetic factors.

Some of them have not been functionally tested and most of them lack large genetic and histological analyses of the clinical material. Therefore, in addition to research on cell models, integrated analysis of several or more potential predictive markers performed on a large group of patients is needed. Additionally, the global integration of genomic and clinical data would further help the identification of potential biomarkers [123]. This would allow for a reliable statistical analysis of the relationship between the presence of mutations in selected genes and the expression of selected proteins with the response to the treatment. Many potential genetic predictive markers are at the same time therapeutic targets as indicated in Table 1, so ideally, the predictive panel would also constitute a panel of alternative therapeutic targets. A predictive toolkit that, in my opinion, is worth further validation would consist of four tests:Targeted sequencing of seven genes, namely: *BRAF*, *NRAS*, *NF1*, *RAC1*, *MAP2K1*, *MAP2K2*, and *MAP3K8* (in tumor or liquid biopsy sample)Copy number analysis of five genes, namely: *PTEN*, *CDKN2A*, *MYC*, *MITF*, and *BRAF* (in tumor or liquid biopsy sample)Protein level analysis of selected growth factors and receptors, namely: HGF, EGFR, PDGFRA, PDGFRB, and IGF1R (in a tumor sample).Optionally: determining HGF protein level in a blood sample and MITF protein expression in a tumor sample.

Based on the results of these tests, a board of specialists, i.e., a medical geneticist, molecular biologist, pathologist, and oncologist, would choose the best treatment option tailored to the individual genetic and protein profile of the patient. The genetic part of the test could also be used for further monitoring of melanoma evolution and adapting, if possible, the therapy to the current tumor genetic profile of a patient.

## Figures and Tables

**Figure 1 cancers-14-02315-f001:**
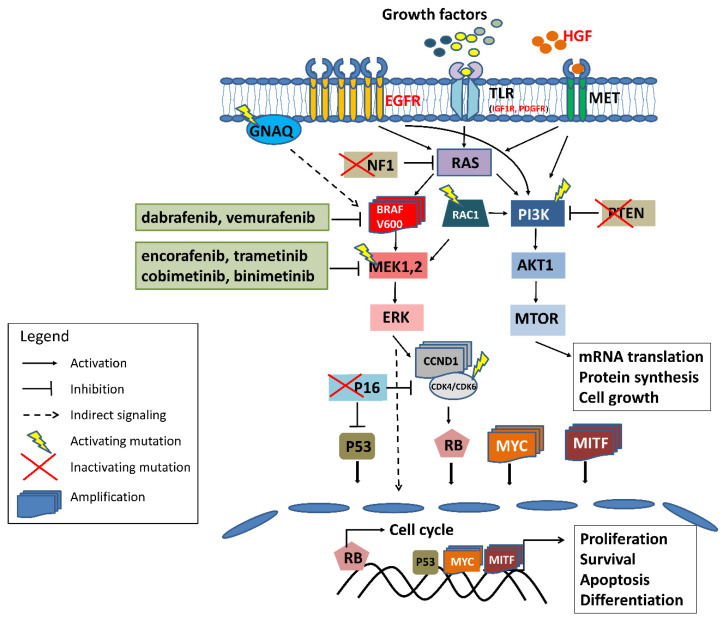
The scheme of the most common and well-established genetic and molecular mechanisms of melanoma resistance to BRAF/MEK inhibitors. Red font denotes overexpression of the protein; in green frames are the inhibitors registered for advanced skin melanoma treatment. For more detailed information refer to the text.

**Table 1 cancers-14-02315-t001:** The most important genetic and non-genetic factors that are involved in melanoma resistance to targeted therapy. The most promising, potential predictive markers are in bold.

Marker	Mechanism of Resistance	Primary vs. Acquired Resistance; Studies	Actionable Mutation/Therapeutic Target	References
**Genetic alterations**				
**NF1**	PI3K/AKT activation	Primary resistance; genetic analysis of samples and functional tests	-	[22,23]
**RAC1** (P29S/L mutation)	PI3K/AKT activation	Primary resistance; occurrence in melanoma samples of patients with early resistance; in vitro and mouse model studies	PAK1 inhibitors	[24,25,26]
**MAP2K1** (mutations: C121S, Q56P, V60E, P124L, G128V, V154L)	MAPK activation	Primary resistance; genetic analysis of samples and in vitro tests	-	[22,24,27]
**MAP2K2** (V35M, L46F, C125S, N126D)	MAPK activation	Primary and acquired resistance; genetic analysis of samples and functional tests	-	[22,24]
**PTEN** (deletions, mutations)	PI3K/AKT activation	Primary resistance modulation; occurrence in pre-treatment melanoma samples in patients with early resistance; in vitro studies	PI3K inhibitors	[22,24]
**CDKN2A**	Deregulation of RB pathway	Primary resistance; genetic analysis of samples	CDK4/6 inhibitors	[24,28,29,30,31]
**MAP3K8** (amplification)	MAPK activation	Primary resistance; genetic analysis of samples and functional tests	-	[32]
**MYC** (amplification, overexpression)	Regulation of proliferation and apoptosis; downstream effector of ERK, PI3K, NOTCH pathways	Primary and acquired resistance; genetic analysis of samples; in vitro studies	Myc inhibition	[28,29,33]
MITF		Primary resistance; genetic analysis of samples		[27]
GNAQ	Unknown	Rapid progression—case report	-	[34]
NRAS	MAPK activation	Mainly acquired resistance; genetic analysis of samples	CDK4/6 inhibitors	[22,24,35]
BRAF (amplification)	MAPK activation	Mainly acquired resistance; genetic analysis of samples	-	[24,36]
BRAF (alternative splicing)	Increased dimerization of short form of BRAF independent of RAS—MAPK activation	Mainly acquired resistance; analysis of samples and functional tests	-	[37,38,39]
**Non-genetic factors**				
**HGF**	Activation of the Met receptor and the MAPK and PI3K/AKT pathways	Primary resistance; analysis of protein and mRNA expression in the tumor and in vitro studies	Drug holidays; Drugs targeting HGF/MET	[40,41,42,43]
**EGFR**	PI3K/AKT activation and MAPK reactivation As above As above	Mainly acquired resistance, possibly also primary resistance; analysis of tumor protein expression	Drug holidays as suggested by Sun [44] et al.	[24,44]
PDGFRB		Mainly acquired resistance; tumor protein expression analysis and in vitro studies		[45]
PDGFRA		Mainly acquired resistance, possibly also primary resistance (as suggested by the authors); in vitro studies		[46]
IGF1R	Inhibition of apoptosis	Mainly acquired resistance; tumor protein expression analysis and in vitro studies	Blocking IGF1R signaling (NT157)	[47,48]
MITF	Low expression induces a phenotype resistant to MAPK inhibitors; high expression induces the expression of anti-apoptotic proteins	Unclear role of MITF expression level in resistance generation		[49,50,51]

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
