# Peer review of "Potential Biomarkers of Skin Melanoma Resistance to Targeted Therapy—Present State and Perspectives"

_cancers, 2022, doi:10.3390/cancers14092315_

Round 1
Reviewer 1 Report
The review article by Magdalena Olbryt is a clear thought-out review of the current field of predictive biomarkers in melanoma. The figure and table particularly will prove useful to those in the field to obtain a summary of the state of the art. I recommend publication of this article, following the addressing of a few minor points as indicated below.
The summary is lacking a bit of an ending. Perhaps it would be useful to summarise the information into a suggested best practice biomarker toolkit- targeted sequencing of gene x y z, determining protein (e.g HGF) levels in the blood and the suggested treatment should be with the results? It’s not clear to me what the point of this review is if not to determine this toolkit?
In section 2. Nice to state what the therapy does, e.g injecting a virus how does that treat the cancer?
More discussion about combinations of mutations which are not favourable for particular drugs?
The figure is also lacking any reference and could be referenced more throughout the text, as could the table.
Below are some minor typos:
BARF typo throughout (go through with FIND and REPLACE)
Genes italicised throughout- mostly correct but a few missed italics
Abstract- up-to-date not day
Table 1 pg 13 inhibitors typo
Section 4.6 frequencies of these mutations
IDTC state what this means-induced drug tolerant cells
Please include line numbers in the future.
Author Response
2nd May 2022
Dear Reviewer,
I would like to thank you for your insightful comments on the manuscript. Below I provide a point-by-point response to the raised queries (in red font).
Major Concerns:
- The summary is lacking a bit of an ending. Perhaps it would be useful to summarise the information into a suggested best practice biomarker toolkit- targeted sequencing of gene x y z, determining protein (e.g HGF) levels in the blood and the suggested treatment should be with the results? It’s not clear to me what the point of this review is if not to determine this toolkit?
Ad.1. Thank you for this valuable suggestion. I have added the proposition of a toolkit at the end of the Summary.
- In section 2. Nice to state what the therapy does, e.g injecting a virus how does that treat the cancer?
Ad. 2. The information was added to the text.
- More discussion about combinations of mutations which are not favourable for particular drugs?
Ad.3. I have extended the discussion about combinations of mutations according to your suggestion and placed the text in a separate chapter.
- The figure is also lacking any reference and could be referenced more throughout the text, as could the table.
Ad.4. The figure was prepared as a summary of the data described in the text and was based on the references placed in the text, not the selected review papers. I referenced the Figure and the Table more throughout the text.
- Minor typos
Ad.5. The mistakes and shortcomings were corrected.
Reviewer 2 Report
The review “Predictive biomarkers in targeted therapy of skin melanoma – present state and perspectives” by Magdalena Olbryt describe the biomarkers, mostly at the same time genes responsible for BRAFi and NRASi resistance. The MS requires some amendments before being suitable for publishing. Some suggestions are described below.
Concerns:
“3. Where does resistance to targeted therapy in melanoma cells come from?”
This is rather short information about the broad problem of drug resistance, which stands close to biomarkers. In fact the name biomarkers and treatment targets can represent the same set of genes, sometimes they are really separated. The statement that “genetic variability of melanoma, resistance to BRAF inhibitors can also be generated by the microenvironment and epigenetic changes” is rather shallow – should be mention that some of the resistance mechanisms are described in the next chapter.
“4. Potential genetic predictive markers in the targeted therapy of melanoma”
This chapter in fact describes (in many subchapters), the mechanisms of resistance to BRAFi and NRASi are described.
In this connection, the terms such as “drug addiction” and “drug holiday”, closely connected to the resistance, are missing and should be added. The is ample evidence and references about these issues. “Drug holiday” is in fact mentioned only once in the Table with connection to EGFR – this is not a prime driver in melanoma, it belongs to the description of BRAFi. In this connection, a paper describing the important involvement of FRA1 and JunB in resistance seems to be missing: Kong X etal: Cancer drug addiction is relayed by an ERK2-dependent phenotype switch. Nature 2017.
“NRAS mutations and BRAF amplification are also often identified in genetic tests.”
This information should be among the first given because BRAF and NRAS constitute the main driving mutations in melanoma, and in other words, not just to say that “are also often identified in genetic tests”.
“5. Potential non-genetic predictive markers in targeted therapy of melanoma”
The description of resistance mechanisms continues, by describing HGF, RTKs,…, and their detailed role in resistance.
Overall: the paper is called “ Predictive biomarkers in targeted therapy of skin melanoma – present state and perspectives” but predominantly deals with mechanisms of resistance to BRAFi and NRASi in melanoma. I understand that these two namings can not be in most cases separated (also above), but the paper title should reflect the content. So, please think about slight modification of the title that would stick more to the content. Factually, the details in descriptions of resistance are correct.
---please add some important missing issues as above ---
Minor concerns:
Simple summary:
Typo: BARF/MEK
-- please correct --
“… most promising predictive and therapeutic factors is presented.”
---did you mean therapeutic targets? ----
Abstract:
“Melanoma is one of the most aggressive skin cancers”
--please delete -one of- and correct to:….cancer. Melanoma is without any doubt the most malignant skin ca.
when V600E mutation is mentioned (and other mutations such as in RAC1, MAP2K1, MAP2K2, NF1), NRAS mutations should be mentioned in the abstract as well, as they represent the second most frequent mutation (present in about 15% of melanomas), after BRAFmut.
- Introduction
-- please add-- Also, it is better to write BRAFV600E throughout the article, as it is the most prevalent BRAF mutation. You could perhaps mention much less frequent BRAF mutations.
- Molecularly targeted therapy
Besides BRAFi/MEKi, you should also mention ERK inhibitor(s) or MEK/ERKi (eg. selumetinib).
Primary or acquired resistance to BRAFi or NRASi occurs in nearly 100% during monotherapies.
FIGURE 1: If inhibitors of BRAF and MEK are shown, there should be also depicted inhibitors of ERK1/2. Also some new specific inhibitors of NRAS are already available.
---Please improve the picture -----
Author Response
2nd May 2022
Dear Reviewer,
I would like to thank you for your insightful comments on the manuscript. Below I provide a point-by-point response (in red font) to the raised queries.
“3. Where does resistance to targeted therapy in melanoma cells come from?”
This is rather short information about the broad problem of drug resistance, which stands close to biomarkers. In fact the name biomarkers and treatment targets can represent the same set of genes, sometimes they are really separated. The statement that “genetic variability of melanoma, resistance to BRAF inhibitors can also be generated by the microenvironment and epigenetic changes” is rather shallow – should be mention that some of the resistance mechanisms are described in the next chapter.
The information was added to the text.
“4. Potential genetic predictive markers in the targeted therapy of melanoma”
This chapter in fact describes (in many subchapters), the mechanisms of resistance to BRAFi and NRASi are described.
In this connection, the terms such as “drug addiction” and “drug holiday”, closely connected to the resistance, are missing and should be added. The is ample evidence and references about these issues. “Drug holiday” is in fact mentioned only once in the Table with connection to EGFR – this is not a prime driver in melanoma, it belongs to the description of BRAFi. In this connection, a paper describing the important involvement of FRA1 and JunB in resistance seems to be missing: Kong X etal: Cancer drug addiction is relayed by an ERK2-dependent phenotype switch. Nature 2017.
I agree that the phenomenon of ”drug addiction” and “drug holiday” is closely connected to resistance, that is why I added the missing information concerning the proteins which are involved in this process. I also included the suggested reference.
“NRAS mutations and BRAF amplification are also often identified in genetic tests.”
This information should be among the first given because BRAF and NRAS constitute the main driving mutations in melanoma, and in other words, not just to say that “are also often identified in genetic tests”.
I agree that BRAF and NRAS are the two most frequently mutated genes in melanoma. However, in my manuscript, I concentrate mainly on the potential primary resistance biomarkers (genes or proteins) which can be detected before therapy in BRAF positive patients and serve as predictive markers. NRAS is detected mainly in progressed samples and is responsible for acquired resistance, similarly to BRAF amplification. That is why these aberrations are described at the end of section 4.
“5. Potential non-genetic predictive markers in targeted therapy of melanoma”
The description of resistance mechanisms continues, by describing HGF, RTKs,…, and their detailed role in resistance.
Overall: the paper is called “ Predictive biomarkers in targeted therapy of skin melanoma – present state and perspectives” but predominantly deals with mechanisms of resistance to BRAFi and NRASi in melanoma. I understand that these two namings can not be in most cases separated (also above), but the paper title should reflect the content. So, please think about slight modification of the title that would stick more to the content. Factually, the details in descriptions of resistance are correct.
I have changed the title according to the suggestion.
---please add some important missing issues as above ---
Minor concerns:
Simple summary:
Typo: BARF/MEK
-- please correct --
“… most promising predictive and therapeutic factors is presented.”
---did you mean therapeutic targets? ----
Abstract:
“Melanoma is one of the most aggressive skin cancers”
--please delete -one of- and correct to:….cancer. Melanoma is without any doubt the most malignant skin ca.
when V600E mutation is mentioned (and other mutations such as in RAC1, MAP2K1, MAP2K2, NF1), NRAS mutations should be mentioned in the abstract as well, as they represent the second most frequent mutation (present in about 15% of melanomas), after BRAFmut.
- Introduction
-- please add-- Also, it is better to write BRAFV600E throughout the article, as it is the most prevalent BRAF mutation. You could perhaps mention much less frequent BRAF mutations.
All the above suggestions were followed and the corrections were made.
- Molecularly targeted therapy
Besides BRAFi/MEKi, you should also mention ERK inhibitor(s) or MEK/ERKi (eg. selumetinib).
As I described only already registered targeted drugs for advanced skin melanoma treatment I decided not to expand the list of drugs.
Primary or acquired resistance to BRAFi or NRASi occurs in nearly 100% during monotherapies.
This information was added to the text.
FIGURE 1: If inhibitors of BRAF and MEK are shown, there should be also depicted inhibitors of ERK1/2. Also some new specific inhibitors of NRAS are already available.
---Please improve the picture -----
In both, the text and the Figure, I described only already registered targeted drugs for advanced skin melanoma treatment.
Round 2
Reviewer 1 Report
Most of the editions look good. I am happy to recommend publication.
I particularly liked the toolkit addition!
Double check for typos e.g GANQ in section 4.7
Reviewer 2 Report
The revised review newly entitled “Potential biomarkers of skin melanoma resistance to targeted therapy– present state and perspectives” by Magdalena Olbryt describe the biomarkers and some mechanisms responsible for BRAFi and NRASi resistance. The author improved the paper sufficiently to be suitable for publishing. The concerns raised in v1 of the paper were addressed or explained, the paper is now more consistent and balanced.